# Relationship between Selected Cephalometric Parameters, Nasolabial Angle and Its Components in Adolescent Females

**DOI:** 10.3390/diagnostics13061199

**Published:** 2023-03-22

**Authors:** Mikołaj Gołębiowski, Agnieszka Świątkowska, Przemysław Pastuszak, Mansur Rahnama

**Affiliations:** 1Department of Jaw Orthopedics, Medical University of Lublin, 20-093 Lublin, Poland; agnieszka.swiatkowska@umlub.pl (A.Ś.); przemyslawpastuszak@umlub.pl (P.P.); 2Department of Oral Surgery, Medical University of Lublin, 20-093 Lublin, Poland; mansur.rahnama@umlub.pl

**Keywords:** nasolabial angle, inclination, upper incisors, upper lip thickness, cephalometry, nasal tip morphology

## Abstract

Nasolabial angle is commonly used to assess the soft tissue profile of the subnasal region. The aim of this retrospective study was to evaluate the relationship between the nasolabial angle, the inclination of the lower border of the nose and upper lip, upper incisor inclination and upper lip thickness. A sample of 142 female adolescents aged 13–18 years was chosen. A modified cephalometric analysis was performed with the nasolabial angle, and its components were traced according to Fitzgerald’s method. All analysed parameters showed a statistically significant correlation with the nasolabial angle (NLA). The highest correlation was found for the labial (L/FH) and nasal (N/FH) components of the nasolabial angle, respectively. Upper incisor inclinations (1+:SN, U1FA) and upper lip thickness (ULT) had a stronger correlation with L/FH than NLA, but no correlation was found between these parameters and N/FH. Upper lip thickness did not influence the relationship between incisor inclination and NLA or L/FH. The position of the upper incisors and upper lip thickness influence the nasolabial angle indirectly through its labial component (L/FH). Therefore, it seems purposeful to assess the nasolabial angle as a sum of two independent angles, of which only one (L/FH) can be influenced by orthodontic treatment.

## 1. Introduction

The desire to improve facial appearance and smile aesthetics is one of the main reasons that motivate patients to seek orthodontic treatment [1,2,3,4]. It is widely accepted that a disturbance of facial aesthetics caused by malocclusion can have a significant negative impact on a person’s self-esteem, as well as physical well-being. Improvement of one’s appearance as a result of orthodontic treatment may have a positive influence on their social and mental status. Modern orthodontic diagnostics should therefore consider not only the skeletal and dental relationships but also the patient’s soft tissue profile. When planning the treatment, orthodontists should pursue methods allowing correction of the malocclusion along with maintaining or, preferably, even improving facial harmony.

One of the most valuable parameters used in soft tissue profile analysis is the nasolabial angle (NLA), formed by the intersection of the upper lip anterior and columella at the subnasale [5]. The mean value of NLA is 102° +/− 8° [5,6,7]. However, the measurement by itself does not provide adequate information about the component responsible for the variability [8,9]. A patient with retruded upper incisors, a receding upper lip and a drooping nose may have a nasolabial angle within the norm despite unsatisfactory subnasal aesthetics. Factors that may influence the nasolabial angle are upper incisor position [10,11,12], upper lip thickness [10], and the inclination of the lower border of the nose and upper lip [8]. In light of these reflections, it seems crucial to analyse each of the nasolabial components separately in order to avoid a diagnostic error that may negatively influence the treatment results.

This retrospective study was designed to carefully identify factors contributing to the value of the nasolabial angle as a means to aid clinicians in better understanding the complexity of the soft tissues in the subnasal region. The relationships between nasolabial angle (NLA), the inclination of the lower border of the nose (N/FH) and upper lip (L/FH), upper incisor inclination (U1FA, 1+:SN) and upper lip thickness (ULT) were investigated.

## 2. Materials and Methods

This study used lateral radiographs of patients treated in the Orthodontic Department of the Medical University of Lublin. The sample consisted of 142 adolescent female patients with the X-ray taken between the age of 13 and 18. Subjects with various configurations of nose and lip profiles were chosen, including down- and upturned noses, recessed lips, protruded lips and combinations of these. All cephalometric X-rays were taken with the lips in a relaxed position. The exclusion criteria were: the presence of congenital facial deformities, brackets bonded to anterior teeth and low X-ray image quality. The sample included patients both pre- and posttreatment, as a history of orthodontic intervention did not influence the results. As the radiographs were taken prior to the undertaking of this study and were analysed retrospectively, no approval from the Ethics Committee was required.

The X-rays were digitalised and added to Orthodontics 9 software—a commercial database used in our University Orthodontic Department. An individualised cephalometric analysis, designed specifically for this study, was performed by an orthodontic resident who was pre-trained by an experienced supervisor teacher. The following nine reference points were marked:Sella—SNasion—NPorion—PoOrbitale—OrIncisal edge of the upper central incisor—IssRoot apex of the upper central incisor—IsaLabrale superius—LsPosterior columella—PCmColumella—ctg

Five angular measurements were recorded using these reference points:Nasolabial angle (NLA)Upper incisor to sella-nasion line (1+:SN)Upper incisor to Frankfort horizontal plane (U1FA)Upper lip to Frankfort horizontal plane (L/FH)Lower border of the nose to Frankfort horizontal plane (N/FH)

Because of the diverse anatomy of the subnasal region and numerous methods of drawing the nasolabial angle, a systematised approach was needed to trace the reference points responsible for its determination. To achieve this goal, we used a three-step approach described in detail by Fitzgerald et al. [8]:(1)The most posterior point of the lower border of the nose, at which it begins to turn inferiorly to merge with the philtrum of the upper lip, was located and called the posterior columella point, or PCm.(2)A tangent was drawn from PCm anteriorly along the lower border of the nose at its approximate middle third; the angle of this line extending anteriorly and intersecting the Frankfort horizontal plane was considered the relative inclination of the nose and termed lower nose to FH angle, or N/FH (nasal component of the nasolabial angle). If this line representing the lower border of the nose was parallel to the Frankfort horizontal plane, it was measured at 0°. In some cases, the tip of the nose was turned so far down that the tangent to the lower border of the nose intersected the Frankfort horizontal plane posteriorly. In this instance, a negative value of the anteroinferior angle (N/FH) formed at this intersection was recorded.(3)A line drawn from PCm to the labrale superius (Ls), when extended superiorly, intersects the Frankfort Horizontal plane and forms an angle considered the relative inclination of the upper lip and termed upper lip to FH angle, or L/FH (the labial component of the nasolabial angle).

The nasolabial angle (NLA) is formed at the intersection of the upper lip and lower border of the nose tangent, and its value is equal to the sum of N/FH and L/FH angles. The above-mentioned angles were jointly termed “nasolabial parameters”.

The inclination of the upper incisors was measured using two reference planes: Frankfort horizontal (U1FA) and SN line representing the base of the skull (1+:SN). These angles were jointly termed “upper incisor inclinations”.

The upper lip thickness measurement was carried out according to the method described by Asmar [13] and Bergman [14]—from the labrale superius (Ls) point to the inside of the lip where the maxillary incisor rests.

All of the angular measurements are presented in Figure 1.

The obtained data were subjected to statistical analysis. A significance level of *p* < 0.05 was established, indicating the presence of statistically significant differences or relationships. The database and statistical analysis were performed using Statistica 9.1 (Statsoft, Poland) and PQStat 1.8.4 software.

In order to verify the measurement reliability, all cephalometric analyses were performed twice by the same person. No statistically significant differences were found between the two measurements. The mean and standard deviation were calculated for each variable using the Shapiro–Wilk test. In order to determine the linear correlation between the examined parameters, r-Pearson correlation coefficients were calculated.

Patients were divided into three groups to verify whether upper lip thickness (ULT) influences the relationship between incisor inclinations and nasolabial parameters:(1)Patients with thin upper lip—ULT value between the minimum and first quartile (*n* = 36)(2)Patients with normal upper lip—ULT value between the first and third quartile (*n* = 70)(3)Patients with thick upper lip—ULT value between the third quartile and maximum (*n* = 36)

The linear correlation coefficients for each group were calculated between upper incisor inclinations, upper lip thickness and the nasolabial parameters. Absolute values of correlation coefficients were also compared with each other to determine any statistically significant differences.

## 3. Results

Table 1 shows the mean, standard deviation and range for all analysed variables.

All variables showed a statistically significant correlation with nasolabial angle. The strongest correlation was observed for L/FH (r = 0.715, *p* < 0.001) and N/FH angles (r = 0.574, *p* < 0.001), while the weakest was for upper lip thickness (r = −0.199, *p* = 0.018). Values of the r-Pearson correlation coefficients are presented in Table 2.

We compared the absolute values of the correlation coefficients of the above-mentioned variables with each other. A statistically significant difference in the magnitude of influence on the nasolabial angle was found between:(1)L/FH and U1FA (*p* < 0.001)(2)L/FH and 1+:SN (*p* < 0.001)(3)L/FH and ULT (*p* < 0.001)(4)L/FH and N/FH (*p* = 0.043)(5)N/FH and ULT (*p* < 0.001)(6)N/FH and U1FA (*p* = 0.028)(7)N/FH and 1+:SN (*p* = 0.042)

No statistically significant difference in correlation with nasolabial angle was found between:(1)1+:SN and U1FA (*p* = 0.869)(2)U1FA and ULT (*p* = 0.121)(3)1+:SN and ULT (*p* = 0.086)

A comparison of linear correlation coefficients with each other is presented in Table 3.

Subsequently, patients were divided into three groups differing in upper lip thickness in order to establish whether it influenced the correlation between the nasolabial parameters and the other variables.

Upper incisor inclinations (U1FA and 1+:SN) showed a strong and statistically significant negative correlation with NLA in all of the groups. Upper lip thickness was not significantly correlated with nasolabial angle in any of the groups, despite the presence of a significant correlation with the entire sample (Table 4).

The correlation between incisor inclinations, upper lip thickness and nasolabial angle components separately was also evaluated for the entire sample and each of the different ULT groups. As expected, no significant correlation was found between incisor inclinations (U1FA, 1+:SN), upper lip thickness and the nasal component (N/FH).

As the labial component (L/FH) showed the strongest correlation with the nasolabial angle, the authors focused on establishing detailed relationships between L/FH and incisor inclination (U1FA, 1+:SN) and upper lip thickness (ULT). All of these variables had a significant (*p* < 0.001) correlation with the labial component (L/FH) for the entire sample. After dividing the patients into groups differing in ULT, both variables describing incisor inclination had a significant (*p* < 0.001) correlation with L/FH. In order to verify whether the correlation was stronger in any of the groups, a mutual comparison of coefficients was carried out. Only one significant (*p* = 0.036) difference was identified: between 1+:SN in patients with thin (r = −0.744, *p* < 0.001) and normal lips (r = −0.467, *p* < 0.001). This finding suggests that incisor inclination influences the labial component of the nasolabial angle more in patients with a thin upper lip compared to those who have an average lip thickness.

The correlation between upper lip thickness and labial component (L/FH) was found to be significant only in patients with a thick upper lip (r = −0.451, *p* = 0.006).

It is worth emphasising, that upper lip thickness showed a notably stronger correlation with L/FH (r = −0.334, *p* < 0.001) than with the nasolabial angle (r = −0.199, *p* = 0.018) for the entire sample. The results of the above-mentioned analysis are presented in Table 5.

Mutual comparison of linear correlation coefficients of incisor inclinations (U1FA, 1+:SN) and upper lip thickness (ULT) with the labial component (L/FH) for the entire sample showed no significant differences (Table 6). It shows that these parameters similarly influence the L/FH angle, thus indirectly modifying the nasolabial angle itself.

Partial correlation coefficients were also calculated between incisor inclinations (U1FA, 1+:SN) and the nasolabial angle (NLA), and then its nasal and labial components separately. This allowed us to verify the linear relationship between these parameters, which was additionally modified by the upper lip thickness (ULT) value. Incisor inclinations were significantly correlated with the nasolabial angle (NLA) and its labial component (L/FH), while no significant correlation was found for the nasal component N/FH, which further strengthens our findings. The results were very similar for both parameters describing incisor inclination. The results of this analysis are presented in Table 7 and Table 8, respectively.

## 4. Discussion

Clinical experience has led authors to analyse parameters that may influence the nasolabial angle and its components. The nasolabial angle tracing method was based on a similar study by Fitzgerald et al. [8]. They analysed the mutual relationship within the nasolabial parameters and their correlation with six skeletal measurements. However, incisor inclination and upper lip thickness were not taken into consideration. In their study, the nasal component N/FH showed the strongest correlation with nasolabial angle—N/FH (r = 0.685, *p* < 0.001) vs. L/FH (r = 0.594, *p* < 0.001). In our study, we found the correlation between L/FH and NLA to be stronger than N/FH and NLA, and the difference was statistically significant (*p* = 0.043).

Nasal and labial components’ correlation coefficients with nasolabial angle were significantly stronger than those of other variables. This observation shows that the soft tissue configuration of the upper lip and nose is mostly responsible for the nasolabial angle value.

Kuhn et al. [15] analysed the influence of saggital change in maxillary incisor position on an angle similar to L/FH, formed between Frankfort horizontal and upper lip tangent (Ls-Sn), called upper lip inclination angle or epsilon (ε). Horizontal changes were measured using three landmarks on the labial surface of the maxillary incisor. They reported a significant linear correlation of the epsilon (ε) angle with both retraction and protraction of upper incisors. The correlation was strongest for the most anterior point of the maxillary incisor. In contrast to our study, the upper incisor position was expressed as a distance rather than an angle.

Lo and Hunter’s [11] study examined the impact of incisor retraction in subjects with Class II, Division 1 treated both with and without extractions on the nasolabial angle in regards to the nasal and labial components separately. They revealed that 90% of the nasolabial angle (NLA) increase was due to the upper lip retraction, while 10% was related to the change of inclination of the lower border of the nose. Incisor position was assessed using upper incisor to Frankfort horizontal angle (UIFA), while Frankfort nasal angle (FNA) and Frankfort labial angle (FLA), similar to N/FH and L/FH used in our study, were the two components of the nasolabial angle. However, there was a difference in the angle drawing method, as in their study, the labial and nasal tangents intersected at Subnasale instead of the posterior columella (PCm).

Following the division of patients into groups differing in upper lip thickness, incisor inclinations were significantly correlated with nasolabial angle and its labial component (L/FH) in all of the groups.

Talass et al. [10] analysed the changes in soft tissue profile following the retraction of maxillary incisors in females with class II division 1 malocclusion and observed a higher posttreatment increase of the nasolabial angle in patients with a thin upper lip. However, the amount of upper lip retraction was larger in subjects with a thicker upper lip before treatment. It is in accordance with the findings of Hodges et al. [16], who examined the response of both upper and lower lip to incisor retraction after extraction of four first premolars and reported a bigger change in their position in patients with thick lips. Other studies [9,17,18,19,20,21,22,23,24] reported that a thick upper lip is less sensitive to changes in incisor position than a thin lip.

Alkadhi et al. [24] analysed the role of upper lip thickness in response to saggital change of maxillary incisor position following extraction of two upper premolars and reported no significant differences in the amount of upper lip retraction in patients with different lip thicknesses.

In our study, no statistically significant correlations were observed between incisor inclinations (U1FA, 1+:SN), upper lip thickness and the nasal component of nasolabial angle (N/FH). This observation indicates that the inclination of the lower border of the nose is independent of the position of the upper lip and incisors. Thus, one should not expect to achieve its notable change during the course of orthodontic treatment. The anatomical structures responsible for supporting the columellar base are the caudal septum, nasal spine and the medial crura of the lower lateral cartilages [25,26]. Patients who suffer from unsatisfactory nasolabial aesthetics caused by an upturned or droopy nose should therefore seek help with plastic surgery procedures such as surgical rhinoplasty [25], nonsurgical rhinoplasty using hyaluronic acid [27,28,29], calcium hydroxylapatite [29] and Aptos threads nose lifting [30,31].

Upper lip thickness showed a weak yet significant correlation with the nasolabial angle. Both incisor inclination parameters had a stronger correlation with NLA than ULT did, but the difference was not statistically significant. A similar situation was observed when analysing the relationship of these variables with the L/FH angle.

## 5. Limitations

The authors did not consider additional soft tissue features such as lip competence and lip strain, which could potentially influence the relationship between incisor inclination and the nasolabial parameters. Some studies underline that these factors may be clinically relevant [12,15,16,17,21].

In most of the cited studies, a different method of drawing the nasolabial angle was used, where the upper lip and nose tangent intersected at the subnasale. This could have potentially made comparing results more difficult.

The number of patients in groups with thin and thick upper lips was considerably small (*n* = 36), which could have potentially made observing significant correlations harder.

Lastly, all of the data collection and cephalometric analyses were carried out by one examiner who was not blinded to the data, which could introduce a risk of bias in this study.

## 6. Conclusions

The findings of this retrospective investigation of the relationships of tissues in the nasolabial region are as follows:The strongest correlation with the nasolabial angle was observed for its labial component—L/FH angleThe L/FH angle is significantly influenced by incisor inclinations (U1FA, 1+:SN) and upper lip thickness.The weakest yet still a significant correlation between the nasolabial angle and L/FH angle was found for upper lip thickness.None of the analysed variables showed a significant correlation with the nasal component of the nasolabial angle, which indicates that the columellar base described by the N/FH angle does not depend on the position of the upper lip or maxillary incisors.

These conclusions lead authors to suggest the following:In order to comprehensively analyse the subnasal region and avoid a diagnostic error, it seems purposeful to measure the nasolabial angle according to Fitzgerald’s [8] method and consider it as a sum of two independent angles, of which only one can be notably altered through orthodontic therapyFurther research is required to investigate the influence of different treatment methods on each of the nasolabial angle components separately.

## Figures and Tables

**Figure 1 diagnostics-13-01199-f001:**
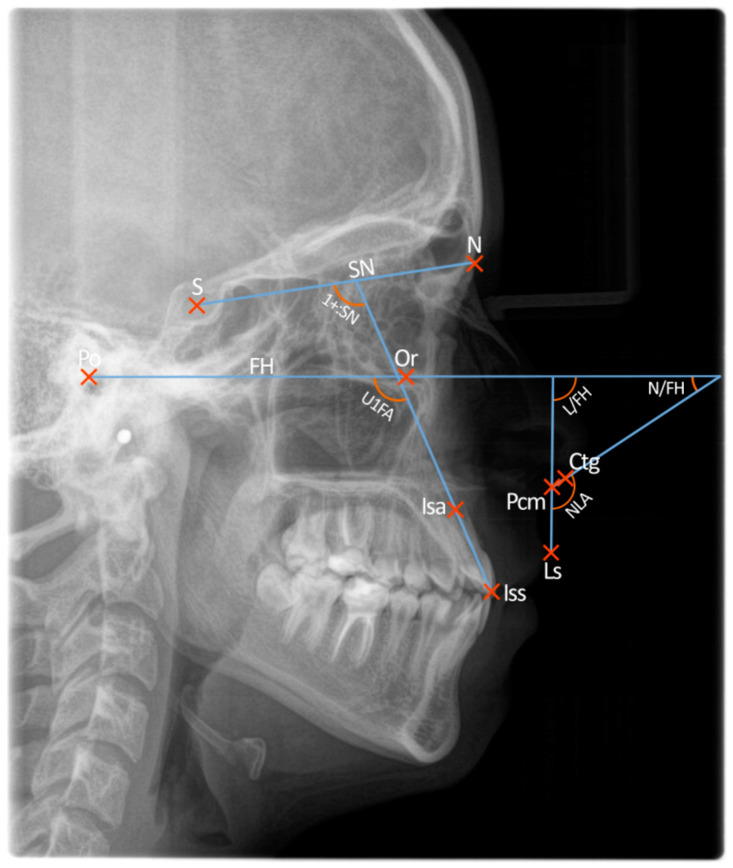
Analysed angular measurements: NLA, N/FH, L/FH, U1FA, 1+:SN.

**Table 1 diagnostics-13-01199-t001:** Mean, standard deviation and range for all analysed parameters.

Variable	Mean and SD	Range
U1FA	112.79° ± 9.52°	92.4°–139.8°
1+:SN	104.17° ± 9.84°	85.4°–131.5°
NLA	114.07° ± 10.81°	86.7°–138°
N/FH	20.3° ± 7.65°	0.3°–37.8°
L/FH	93.77° ± 8.97°	57.6°–117.6°
ULT	12.04 mm ± 1.96 mm	7 mm–16.9 mm

**Table 2 diagnostics-13-01199-t002:** Linear correlation coefficients between the nasolabial angle and the other variables.

Analysed Variable	Correlation with Nasolabial Angle	
	r	*p*
U1FA	−0.370	<0.001
1+:SN	−0.387	<0.001
N/FH	0.574	<0.001
L/FH	0.715	<0.001
ULT [mm]	−0.199	0.018

**Table 3 diagnostics-13-01199-t003:** Correlation coefficients of analysed variables with NLA and their mutual comparison.

Analysed Variable	Correlation with Nasolabial Angle		Significance of Comparison			
	r	*p*	1+:SN	N/FH	L/FH	ULT
U1FA	−0.370	<0.001	0.869	0.028	<0.001	0.121
1+:SN	−0.387	<0.001	–	0.042	<0.001	0.086
N/FH	0.574	<0.001	–	–	0.043	<0.001
L/FH	0.715	<0.001	–	–	–	<0.001
ULT [mm]	−0.199	0.018	–	–	–	–

**Table 4 diagnostics-13-01199-t004:** Correlation coefficients of U1FA, 1+:SN and ULT with NLA for groups differing in ULT.

Analysed Variable	Correlation with NLA					
	ULT [mm]: 7.00 ≤ x ≤ 10.50 (*n* = 36)		ULT [mm]: 10.50 < x < 13.23 (*n* = 70)		ULT [mm]: 13.23 ≤ x ≤ 16.90 (*n* = 36)	
	r	*p*	r	*p*	r	*p*
U1FA	−0.435	0.008	−0.494	<0.001	−0.417	0.011
1+:SN	−0.489	0.002	−0.448	<0.001	−0.413	0.012
ULT	0.119	0.489	−0.097	0.423	−0.315	0.061

**Table 5 diagnostics-13-01199-t005:** Correlation coefficients of incisor inclinations (U1FA, 1+:SN)and upper lip thickness (ULT) with L/FH.

Analysed Variable	Correlation with L/FH Angle							
	Entire Sample		I. ULT [mm]: 7.00 ≤ x ≤ 10.50 (*n* = 36)		II. ULT [mm]: 10.50 < x < 13.23 (*n* = 70)		III. ULT [mm]: 13.23 ≤ x ≤ 16.90 (*n* = 36)	
	r	*p*	r	*p*	r	*p*	r	*p*
U1FA	−0.446	<0.001	−0.708	<0.001	−0.564	<0.001	−0.504	<0.001
Coefficients comparison	–		I vs. II: *p* = 0.253, I vs. III: *p* = 0.187, II vs. III: *p* = 0.694					
1+:SN	−0.434	<0.001	−0.744	<0.001	−0.467	<0.001	−0.439	0.007
Coefficients comparison	–		I vs. II: *p* = 0.036, I vs. III: *p* = 0.051, II vs. III: *p* = 0.869					
ULT	−0.334	<0.001	0.103	0.552	−0.131	0.281	−0.451	0.006

**Table 6 diagnostics-13-01199-t006:** Correlation coefficients of L/FH with analysed variables and their mutual comparison.

Analysed Variable	Correlation with L/FH		Significance of Comparison	
	r	*p*	1+:SN	ULT
U1FA	−0.446	<0.001	0.901	0.271
1+:SN	−0.434	<0.001	–	0.328
ULT [mm]	−0.334	<0.001	–	–

**Table 7 diagnostics-13-01199-t007:** Partial correlation coefficients of nasolabial angle (NLA) with U1FA and 1+:SN controlled by upper lip thickness (ULT).

Analysed Variable	Correlation with NLA	
	r	*p*
U1FA	−0.452	<0.001
1+:SN	−0.452	<0.001

**Table 8 diagnostics-13-01199-t008:** Partial correlation coefficients of N/FH and L/FH with U1FA and 1+:SN controlled by upper lip thickness (ULT).

Analysed Variable	Correlation with N/FH		Correlation with L/FH	
	r	*p*	r	*p*
U1FA	0.033	0.698	−0.595	<0.001
1+:SN	−0.014	0.870	−0.554	<0.001

## Data Availability

The data presented in this study are available on request from the corresponding author.

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
