# Peer review of "Relationship between Selected Cephalometric Parameters, Nasolabial Angle and Its Components in Adolescent Females"

_diagnostics, 2023, doi:10.3390/diagnostics13061199_

Round 1
Reviewer 1 Report
The goal of this study was to examine the connections between the nasolabial angle, the inclination of the lower border of the nose and upper lip, the inclination of the upper incisors, and the thickness of the upper lip.
One limitation of using X-rays as a method of measurement in this study is that it only provides a two-dimensional view and may not fully capture the complexities of the relationships being studied. It would be beneficial for the authors to consider using a more comprehensive method of measurement, such as 3D imaging. Additionally, including assessments of intra- and inter-rater reliability would strengthen the findings and increase the confidence in the results. These modifications would enhance the study's contribution to the understanding of facial morphology and inform orthodontic treatment planning.
Reviewer 2 Report
Dear Authors.
Thank you for this interesting paper. describing the relationship between selected cephalometric parameters, na-2 solabial angle and its components in adolescent females.
"Ortobajt software database" should be decribed, is it a commercial database?
Was this research approved by the ethical vommittee of you institution?
Who did the measurements?
Was the measurements verified by a second or third qualified person.
Were the examiners blinded to the other examiners observations?
Was there any pre training of the examiners?
The multiple abbreviations make the paper difficult to read, possibly some simplification or more explanation of the table description will be helpfull.
The research will be more valuable if the post traement scores are also looked at at a later stage.
Round 2
Reviewer 1 Report
Thank you for addressing my comments.
Author Response
Your welcome
Reviewer 2 Report
Dear Authors
Thank you for the improvements.
I would reccomend including a description of the ethicla reuirements regarding this study in the artile as well as mentioning the limitations in the article regarding no blinded evaluation of the data as well as only one examiner.
Kind regards
Author Response
The recommended changes have been made